# A novel experimental design approach to generating orbital angular momentum waves using wearable textile antenna for sub-6 GHz 5G

Shehab Khan Noor[1], Arif Mawardi Ismail[1], Nassrin I. M. Elamin[2]*,
Mohd Najib Mohd Yasin[1], Mohamed Nasrun Osman[1], Ping Jack Soh[3], Nurulazlina Ramli[4],
Ali H. Rambe[5], Adel Y. I. Ashyap[6]

**1** Advanced Communication Engineering (ACE), Centre of Excellence, Faculty Electronic Engineering Technology, University Malaysia Perlis, Kangar, Perlis, Malaysia, **2** Faculty of Electronics and Electrical Engineering, International University of Africa, Khartoum, Sudan, **3** Centre for Wireless Communications (CWC), University of Oulu, Oulu, Finland, **4** Centre for Advanced Electrical & Electronic System (CAEES) Faculty of Engineering, Built Environment and Information Technology, SEGi University Petaling Jaya, Selangor, Malaysia, **5** Advanced Telecommunication Research Center (ATRC), Faculty of Electrical and Electronic Engineering, Universiti Tun Hussein Onn Malaysia (UTHM), Batu Pahat, Johor, Malaysia, **6** Department of Electrical Engineering, Universitas Sumatera Utara, Medan, Indonesia

* nisreenalameen@gmail.com

## Abstract

This paper presents a novel wearable textile array antenna designed to generate Orbital Angular Momentum (OAM) waves with mode +1 at 3.5 GHz (3.4 to 3.6 GHz) of the sub-6 GHz 5G New Radio (NR) band. The proposed antenna is based on a uniform circular array (UCA) of four microstrip patch antennas on a felt textile substrate. Compared to previous works involving the use of hard substrates for OAM waves generation, this work explored the use of flexible textile substrates to generate OAM waves for the first time to the best of our knowledge. The overall dimension of the array antenna is $170 \times 156\,mm^2$ while the dimension of each element is $35 \times 35.7\,mm^2$. In order to control the phase and generate OAM waves, the proposed antenna was designed using a felt textile substrate and meandering lines of various lengths connecting the radiating patches. $1.48\lambda$ was the separation between radiating patches in order to prevent mutual coupling between them. The antenna was fabricated and measured prior to comparison to simulations to validate this feature. It achieved a measured gain of 3.18 dBi with a bandwidth of 430 MHz (3.24 to 3.67 GHz). Additionally, mode purity analysis was carried out to verify the generation of OAM mode +1, and the purity obtained was 52.12%. This paper also covered the effect of bending on OAM waves characteristics and the use of airgap technique to enhance the antenna gain. The antenna gain increased from 3.762 dBi to 5.327 dBi by using 1 mm airgap without affecting the mode purity. Furthermore, as per the Specific Absorption Rate (SAR) obtained, it is found that the proposed antenna is safe for on-body use. The novel approach in generating OAM using patch array antenna with

**Data availability statement:** All relevant data are within the paper.

**Funding:** The author(s) received no specific funding for this work.

**Competing interests:** The authors have declared that no competing interests exist.

flexible substrate by replacing conventional hard substrate has opened up new scope of research in wearable textile antenna domain. The proposed antenna has simple structure, easy to design, fabricate and deploy on human body and has important significance in scaling up this design to generate multiple OAM modes for carrying multiple signals simultaneously.

## 1. Introduction

As the need for high data speed is increasing exponentially in recent years, it is only a matter of time before the current Long-Term Evolution-Advanced (LTE-A) systems achieve its limits. The fifth generation (5G) new radio (NR) system has been developed to solve this problem [1]. It offers data rates that are 100 times faster and latency that is as low as 1 ms. The 5G NR is divided into two main segments, FR1 and FR2, with FR1 covering sub-6 GHz bands (n77, n78, and n79) and FR2 covering millimeter-wave bands [2]. It is necessary to have an efficient spectrum with a high channel capacity to support the need for such a higher data rate for next-generation wireless communication. An OAM wave theoretically has an infinite number of orthogonal modes that are orthogonal to one another. By multiplexing different OAM modes in the same frequency channel, this orthogonality feature presents a unique opportunity to improve spectral efficiency for 5G technology [3]. Furthermore, OAM provides a new "Mode Division Multiple Access" scheme where different mode numbers can be assigned to different users within the same frequency without interfering with one another and without consuming additional resources such as frequency and time [4]. In this way, large amounts of data from different users can be transmitted simultaneously without degrading the system's overall performance. It was observed that OAM has a higher channel capacity than a Multiple-Input-Multiple-Output (MIMO) system under Line-of-Sight (LOS) conditions [5].

There are different methods to generate OAM waves in the radio domain. Nonetheless, the uniform circular array (UCA) regarded as the most mature approach [6]. In UCA, multiple radiating elements are arranged in a circular pattern where the radius from the centre of the antenna to the centre of the radiating elements are same with phase change and equal amplitude among the adjacent elements. Previous works on generating OAM waves involved using rigid substrates such as FR-4, Rogers, etc. However, another less explored potential of such OAM-based communication systems is in the wearable and body-centric communications. Wearable antenna has been investigated and used in a number of fields recently, including security, military, medical sector and defense monitoring [7]. Due to its high flexibility, low weight, and simplicity of integration into clothing, the textile antenna is one of the suitable materials for wearable applications [8]. Despite this, antennas for OAM generation have been mainly implemented on rigid substrates which are uncomfortable to be worn. In [9], an eight-element circular array antenna has been designed on FR-4 substrate to generate OAM waves with mode +1 at 10 GHz. However, gain and mode purity were not studied. Similarly, a ceramic substrate-based UCA composed of six radiating elements was proposed in [10] to generate OAM mode −1 at 1.550 GHz. However, the resulting

antenna size is large, and essential OAM parameters such as mode purity, intensity distribution, and measured phase distribution are not reported. On the other hand, a multi-layered UCA on FR-4 substrate was developed by the authors in [11] to generate Mode +1 and Mode +2 at 2.4 GHz. However, the antenna size was 300 mm × 300 mm, which is too large considering the desired OAM mode numbers. Moreover, the measured operating bandwidth was from 2.26 – 2.35 GHz, and the intensity distribution and mode purity information are not reported. The authors in [12] simulated an eight element triangular patch antenna on a Rogers substrate with relative permittivity of 2.2 and thickness of 1.6 mm to generate OAM mode −1 at 5.45 GHz. However, the phase distribution and radiation pattern were not measured nor the analysis on intensity distribution was performed. A FR-4 substrate based UCA with eight rectangular patch elements was designed and fabricated to generate mode −1 at 10 GHz by the authors in [13]. The authors confirmed the generation of the OAM waves with mode −1 based on the phase distribution and 2D polar form only. The intensity distribution nor the mode purity was analysed in this paper. In order to generate two different OAM modes +1 and −1, a dual port FR-4-based UCA antenna was designed and fabricated to operate from 5.37 to 5.69 GHz in [14]. The overall antenna size was 120 mm × 112 mm × 5.3 mm. A multi-layered UCA antenna on a F4B-2 substrate to generate OAM modes +1 and −1 with fifteen elements [15]. Nevertheless, the authors did not report the information about the intensity distribution nor the mode purity. To generate single OAM mode +1, a circular metasurface based array antenna with circular polarization concept was proposed by the authors in [16] to operate at 3.360 GHz. A total of six elements were used in the array to generate OAM mode +1 however the bandwidth obtained was very narrow. Nevertheless, the authors carried out both the measurements for phase and intensity distribution to confirm the generation of OAM waves. To confirm the successful generation of OAM waves, it is imperative to study and analyze phase distribution [17], intensity distribution [18], 2D radiation pattern [19] and mode purity [20]. Moreover, as reported in [21], one of the challenges in generating OAM waves using UCA is lower antenna gain compared to the other methods. Furthermore, compared to conventional hard substrates, textile substrate-based antennas tend to achieve lower gain. Previously, to increase wearable textile antenna gain, the authors in [22] proposed the concept of reactive impedance substrate (RIS) on felt textile to enhance the gain of their proposed patch antenna at 2.45 GHz. By using the concept of RIS, the antenna gain increased from 4.67 dBi to 6.07 dBi which is 29.97% enhanced gain. In [23], the authors proposed an Ω-shaped design on a felt textile substrate with metamaterial concept to increase the antenna gain. The antenna gain increased from 4.77 dBi to 7.79 dBi which is 63.31% enhancement. However, designing and implementation of metamaterial unit cells is complex and difficult in terms of fabrication. In order to cover multiple frequency bands with wide bandwidth coverage, a FR-4 based triangular monopole antenna with two stubs was designed and fabricated in [24]. By introducing two stubs with p-i-n diodes, the antenna operated from 3.3-4.2 GHz as single band, 3.3- 4.2 GHz and 5.8-7.2 GHz as multi band and 4-7.8 GHz as wide band. To increase the impedance bandwidth of a microstrip patch antenna, the authors in [25] implemented a metamaterial based T-matching network within the feedline. The impedance bandwidth increased by 12% compared to the patch antenna without T-matching network. For an array antenna, coupling between the adjacent radiating elements is a major concern for antenna performance. To mitigate this concern, a decoupling method using U-shaped resonator was proposed by the authors in [26]. By implementing the U-shaped resonator between the two adjacent radiating elements, the isolation between the two radiating elements increased from -9 dB to −23 dB. Another approach to reduce the coupling between the adjacent radiating elements was proposed by the authors in [27] by embedding different lengths of slots near the outer most edge of the radiating patches.

This paper proposes a textile wearable antenna array to generate mode +1 OAM waves. A simple corporate feeding network with its mirror image can be used to generate OAM waves with mode −1. The influence of bending of the human body on the antenna's reflection coefficient and bandwidth was first analysed via measurements and simulations. Besides that, the OAM wave generation from this design was validated by measuring the radiation pattern and phase distribution. Moreover, Specific Absorption Rate (SAR) analysis on the human body was investigated through simulations. A comprehensive investigation of the effect of bending on OAM wave characteristics was performed. Lastly, the implementation of airgap technique was analysed extensively to study its effect on antenna gain, mode purity and bandwidth. It is worth mentioning

that the proposed antenna is the first of its kind in our best knowledge, where OAM waves are generated fully using flexible textile material for 5G NR body-centric communications. The paper is organized as follows. The methodology of the proposed antenna and the features are explained in Section 2. The experimental investigations and analysis of the proposed OAM wearable antenna is summarized Section 3 which include reflection coefficient, bandwidth, surface current, radiation patterns, phase distribution, bending analysis and influence of airgap technique study. Finally, conclusions are drawn in Section 4.

## 2. Methodology of the wearable OAM antenna

### 2.1. Antenna design and fabrication

The configuration of the proposed antenna in simulation and the flexibility of the proposed antenna is shown in Fig 1(A) and Fig 1(B) respectively. It is fabricated on a felt substrate with a relative permittivity of 1.44 and a thickness of 3 mm. The radiating elements and the ground plane were formed using the Shieldit Super Electro-Textile that had a 0.17 mm thickness, and an estimated conductivity ($\sigma$) of $1.18 \times 10^5$ Sm$^{-1}$. The proposed antenna is based on the concept of a uniform circular array, where a total of four identical radiating elements with varying phases were designed. Inset feed technique was used by cutting the two sides of feedline of each element to make sure there is good impedance matching between the transmission line and the patches so that maximum power reaches the patches through the feedline network. For an $N$-element UCA, an OAM wave of topological charge $l$ can be generated by exciting the elements with the power of equal amplitude but with a phase difference of $\Delta\varphi = \frac{2\pi l}{N}$ between adjacent elements [13]. Therefore, to generate OAM mode ($l$) = 1 with four patches, the phase difference has to be 90° between adjacent antenna elements. A carefully designed corporate feed network with the T-junction method was implemented to ensure that fed signal arrives at the patches with equal amplitude and desired phase differences. The combination of 50 $\Omega$ and 100 $\Omega$ feeding network was used to make sure that each radiating elements is receiving equal or almost same power amplitude to generate OAM waves with desired mode number. To avoid mutual coupling issue between the array elements, the spacing between the radiating array elements was 1.48$\lambda$. Fig 1(C) provides a summary of the proposed antenna's fabrication process. First, a piece of paper with the proposed antenna's structure printed with its exact dimensions from the simulation was used as a guide to cut the super-Shieldit electro-textile. Gerber software was used in this stage. The radiating patches and feedline won't bend or shift while being attached to the substrate, which is another benefit of printing the structure with the proper dimensions. In order to physically cut the printed paper with a cutter and scissors, it was tightly fastened to a piece of super-Shieldit fabric. Next, the substrate was cut per the simulated design dimension, and the patches and feedlines were carefully placed on top of the substrate. To excite the antenna, a 50 $\Omega$ female Sub Miniature version A (SMA) connector was soldered onto the feedline. And lastly, a multimeter was used to check the continuity between the SMA connector inner pin and the feedline.

As it can be seen from Fig 2(A), at 3.5 GHz, the phase for patch 1, patch 2, patch 3 and patch 4 were 2°, 91°, 182°, and 273°. The phase difference between patch 1 and patch 2 is 89°, patch 2 and patch 3 is 91°, patch 3 and patch 4 is 91°, and patch 4 and patch 1 is 89°. Based on the calculated value, the phase difference among the adjacent elements should be 90° to generate OAM mode +1, and it has been observed that the obtained values are close to 90°. Besides, the amplitude was almost similar for all the four radiating elements, as illustrated in Fig 2(B). The amplitude for patch 1, patch 2, patch 3 and patch 4, were − 12 dB, − 11 dB, − 12 dB and − 11 dB respectively.

To study the antenna performance when worn, a four layered human phantom was designed for co-simulations in CST software to be analysed based on [28] as illustrated in Fig 3. The human layer sizes were identical to the ground plane size of the proposed OAM wearable antenna (170 mm × 156 mm). The human tissue properties at the desired frequency is tabulated in Table 1.

### 2.2. Antenna bending

Wearable antennas are mainly designed to be mounted on human body. As a result, the bending analysis on the antenna performance such as resonance frequency, bandwidth etc are important. When it comes to OAM based wave antenna, along with conventional antenna performance, it is very important to analyse the bending effect on generated OAM waves

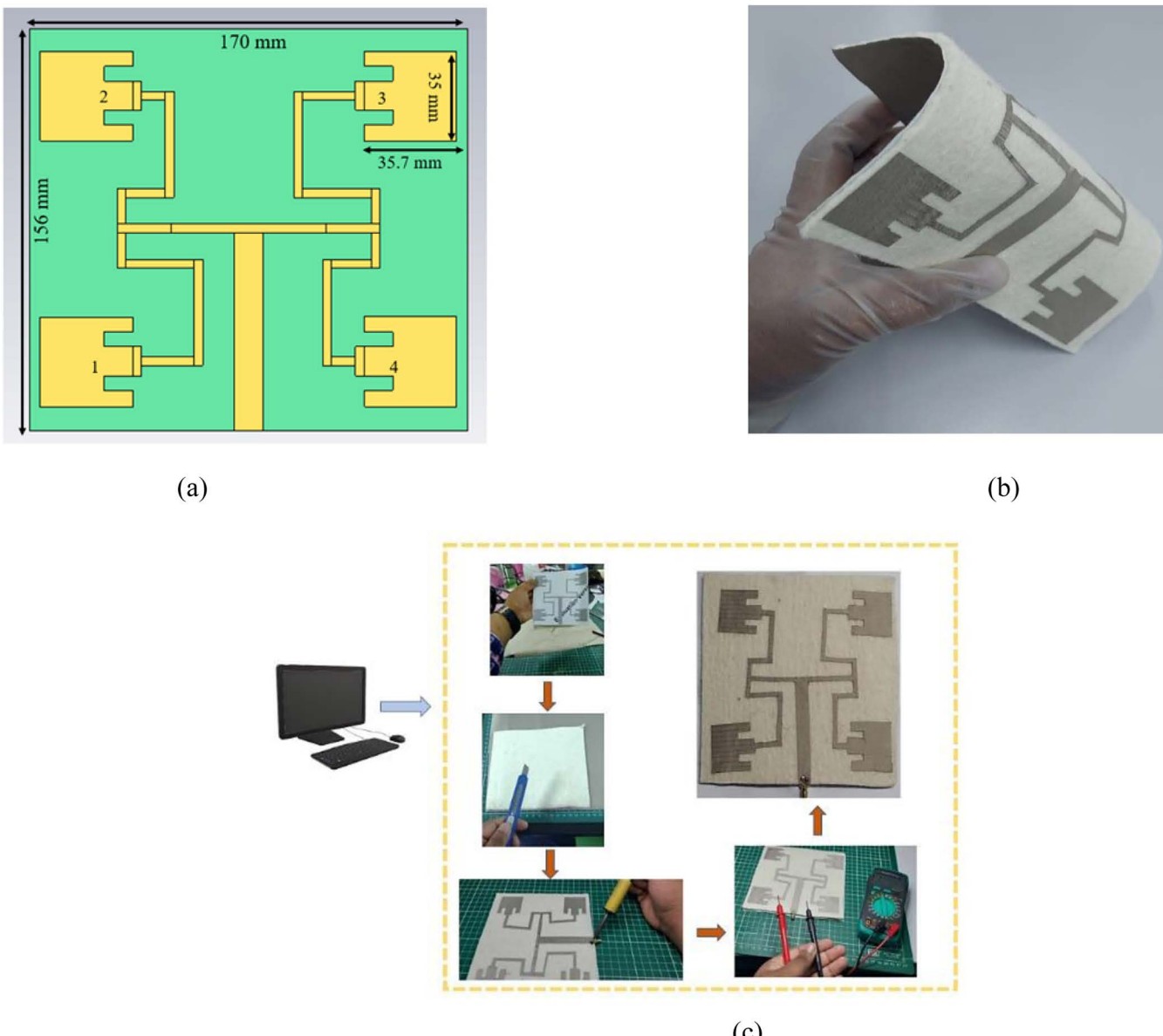

(a)

(b)

(c)

**Fig 1. Configuration of the proposed antenna (A) Simulated design; (B) Flexibility of the proposed antenna; (C) Fabrication process of OAM wearable antenna.**

characteristics. In this paper, four different bending angles are chosen in free space such as 0°, 5°, 10° and 15°. The bending is carried out across the width of the patch as shown in Fig 4. Equation 1 to 3 were used to find the radius of the bending angle. The overall summary of the conversion is tabulated in Table 2.

$$S = \theta \times R \tag{1}$$

$$\theta \ (Radian) = Angle \ in \ degree \times \frac{\pi}{180} \tag{2}$$

$$Radius = \frac{S}{\theta} \tag{3}$$

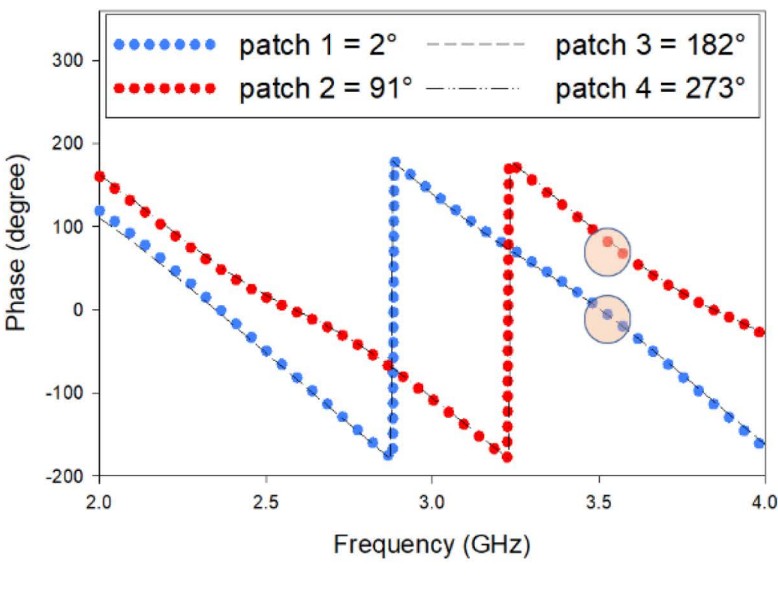

(a)

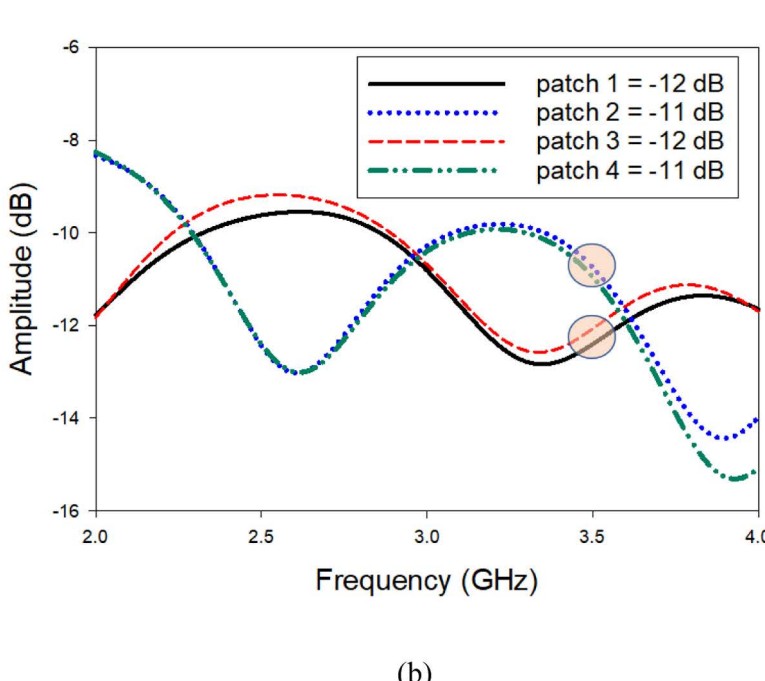

(b)

Fig 2. Simulated (A) Phase and (B) Amplitude of the feed network to generate OAM mode +1.

Here "*S*" is the arc length which is across the width of the antenna of the proposed OAM wearable antenna.

## 3. Results and discussion

### 3.1. Reflection coefficient (S₁₁) and bandwidth

The fabricated antenna's reflection coefficient was simulated using CST Microwave Studio (MWS) Software and measured using a vector network analyzer (VNA) (Agilent E5071C) for five different conditions. The testing of the antenna was

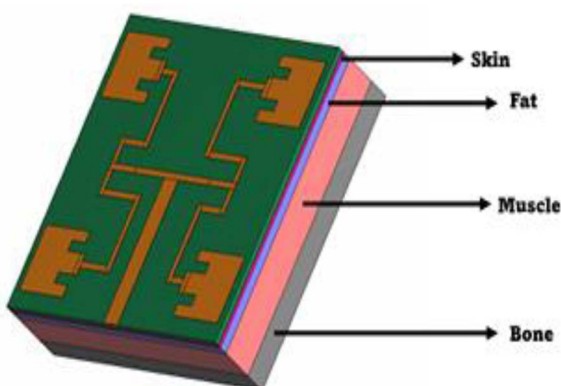

**Fig 3. Simulated four layers human phantom design.**

**Table 1. Properties of human tissue at 3.5 GHz.**

| Tissue | Permittivity ($\varepsilon_r$) | Conductivity (S/m) | Density (kg/m³) | Thickness (mm) |
|---|---|---|---|---|
| Skin | 37.95 | 1.49 | 1001 | 2 |
| Fat | 5.27 | 0.11 | 900 | 5 |
| Muscle | 52.67 | 1.77 | 1006 | 20 |
| Bone | 18.49 | 0.82 | 1008 | 13 |

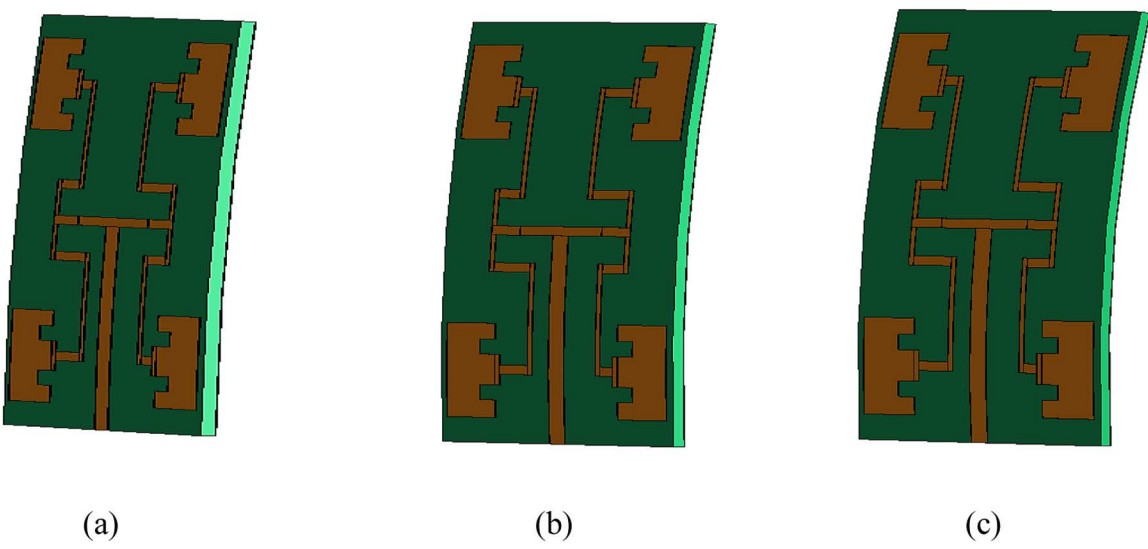

(a) (b) (c)

**Fig 4. Simulated deformation of the proposed OAM wearable antenna (A) 5°; (B) 10°; (C) 15°.**

performed for conditions such as flat free space, on body measurement in flat shape and 5°, 10° and 15° bending in free space. From Fig 5(A), it is observed in all five different conditions, there is minimal shift in terms of operating frequency. In the free space and flat condition, the obtained simulated $S_{11}$ was −10.42 dB at 3.5 GHz, with an overall bandwidth of 932.7 MHz. When its bent, its operating frequency shifted slightly, yet it still operated within the desired frequency range. When

**Table 2. Conversion of angle in degree to radius.**

| Angle (°) | Radian | Radius |
|---|---|---|
| 5 | 0.0872 | 1793.4 mm |
| 10 | 0.1745 | 896.7 mm |
| 15 | 0.261 | 597.8 mm |

assessed on the human phantom in the flat condition, the simulated $S_{11}$ value was −10.47 dB at 3.5 GHz. As from Fig 4(B), in the free space and flat condition, the measured $S_{11}$ value was −19.74 dB, with an overall bandwidth of 430 MHz. The antenna was then bent to 5°, 10° and 15° where 15° was the highest degree of bending resulting in shifts of the operating frequencies towards higher bands. A similar trend was observed when the antenna was placed on the human body. When the fabricated antenna was placed on the human body in flat condition, the $S_{11}$ value at 3.5 GHz was −25.30 dB with an overall bandwidth of approximately 800 MHz. Along with the fabrication error, the impedance mismatch between the radiating elements and the phase divider feeding network may also be responsible for the frequency shift. Besides, the antenna's internal impedance varies when it is placed on a material—in this case, the human body—that has varying dielectric characteristics, causing a shifting in frequency. This effect is called detuning effect. Moreover, the dielectric characteristics of human flesh differ from those of a free space environment and have a high conductivity ($\sigma$), which could have caused the shifting of frequency. In summary, the obtained simulation and measurement results agree well and does not vary significantly despite the different operating conditions, be it in free space and on body, or when assessed in planar condition or bent.

### 3.2. Surface current distribution

The surface current distribution on the proposed antenna at different phase instants at 3.5 GHz is shown in Fig 6. The purpose of analysing the surface current distribution on an antenna is to detect the regions on the antenna where the maximum flow of current is taking place. The maximum surface current obtained at 3.5 GHz was 33.5 A/m. The surface current distribution at phases 0° and 180° are similar as shown in Fig 6(A) and Fig 6(C). At phases 90° and 270° as shown in Fig 6(B) and Fig 6(D), the surface current is identical. When the phase is 0° and 180°, the surface current is strong and identical at the feeds of patches 1 and 3 while at phase 90° and 270°, the surface current is strong and identical at the feeds of patches 2 and 4. Besides, a 90° successive phase delay is also observed between the adjacent elements which contributes to the generation of OAM waves with mode ±1 using four radiating elements.

### 3.3. Radiation pattern and intensity distribution

Fig 7(A) shows the measurement setup in an anechoic chamber. Fig 7(B) presents the simulated and measured results of the normalized radiation pattern obtained for wearable OAM mode +1 which are in good agreement. There is a null at the center of the boresight direction for both simulated and measured 2D polar form which is a characteristic of OAM waves. Moreover, the normalized simulated gain obtained was 3.762 dBi at 340° and the measured gain obtained was 3.18 dBi at 339° with low sidelobes. Nevertheless, there is some unstable side lobes in the measured which is due to the noise equipment during testing the antenna. The radiation measurement operates at 360° and both the Antenna Under Test (AUT) and cables rotate along the process. The vortex core is visible at the center of the simulated intensity distribution plot in Fig 7(C) . The 3D radiation pattern in Fig 7(D) shows null or dip center which is due to the phase singularity on the beam axis.

### 3.4. Phase distribution

For experimental verification, the near-field performance of the proposed antenna was measured in an anechoic chamber, as shown in Fig 8(A). The measurement system mainly consists of the antenna prototype under test (AUT) and a standard gain horn antenna (A-INFOMW LB-20200-SF) mounted on a linear rail that can move horizontally and vertically.

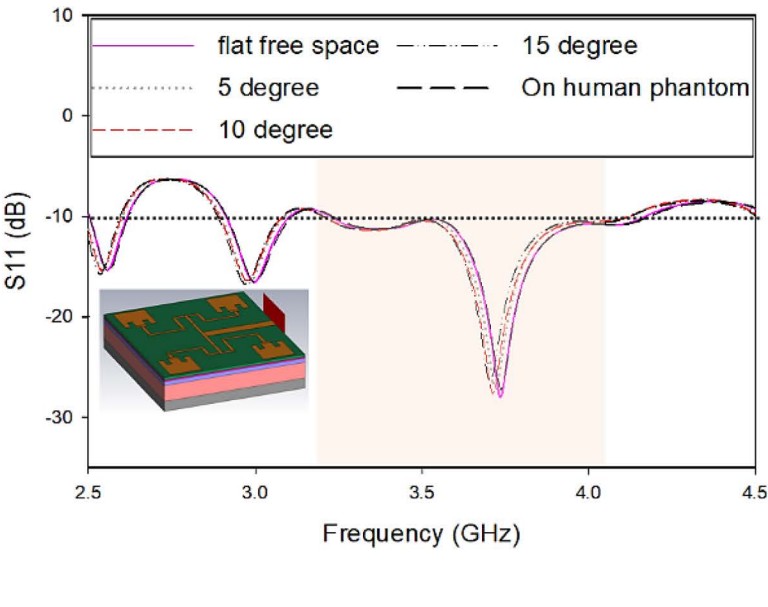

(a)

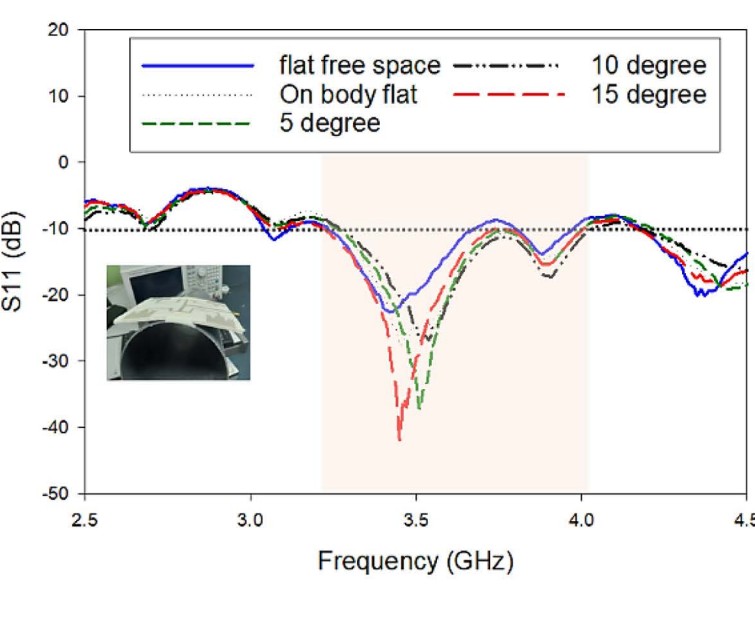

(b)

**Fig 5. S$_{11}$ and Bandwidth of the proposed antenna (A) Simulated; (B) Measured.**

The position of the receiving horn antenna on the linear guide rail was controlled using a computer with a step size of 1 mm. The scanning plane of 200 mm × 200 mm is located 1 m from the proposed antenna. The receiving horn antenna scans the phase of the radiation emitted by the proposed antenna (AUT). Next, the signal is sent to the VNA and the data is stored as in the computer. Lastly, the phase distribution is obtained using Python programming. Despite experimental

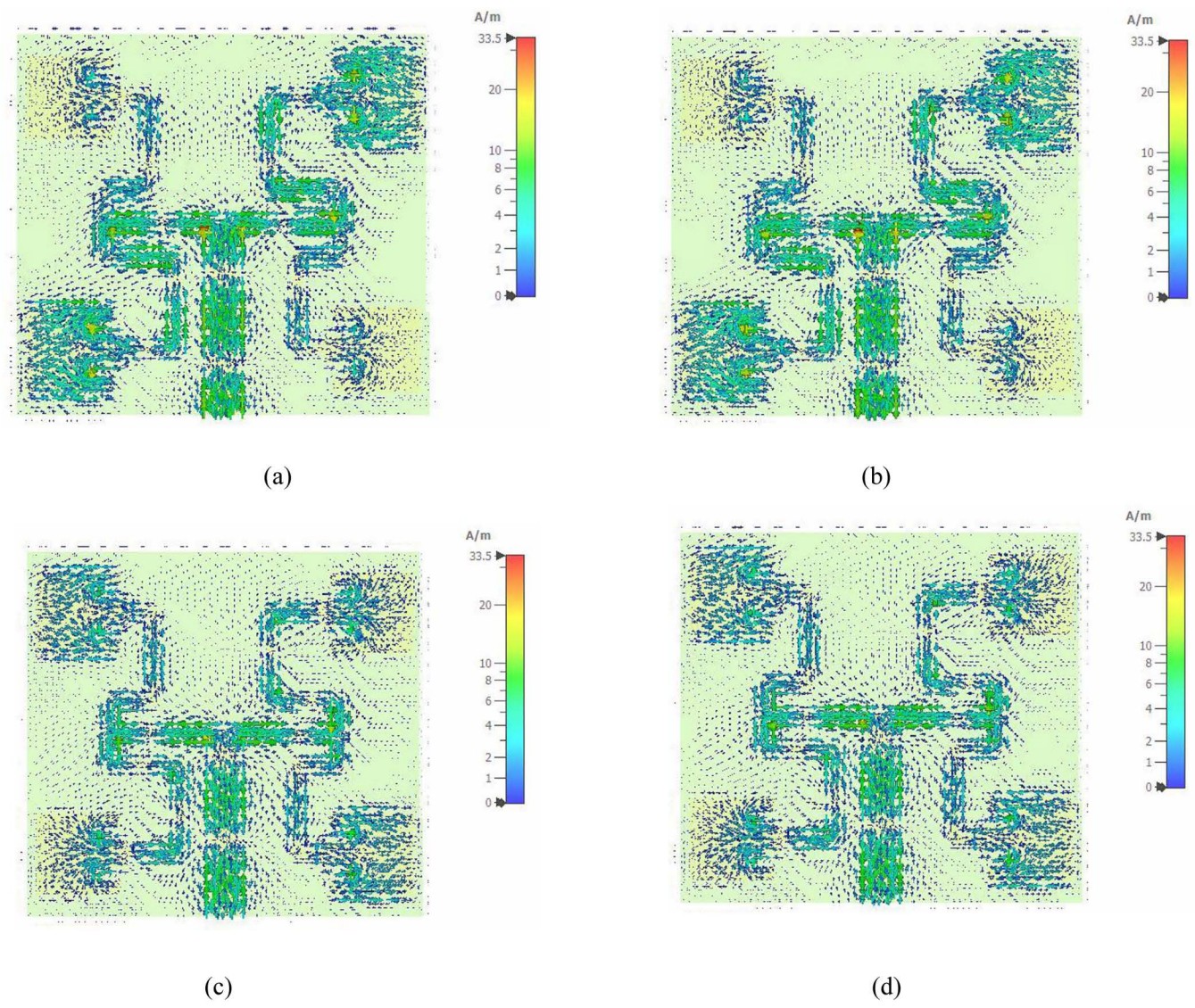

(a)

(b)

(c)

(d)

**Fig 6. Surface current (A) Phase 0°; (B) Phase 90°; (C) Phase 180°; (D) Phase 270°.**

non-idealities, a clockwise spiral phase distribution was observed for both simulation and measurement results, as shown in Fig 8(B) and Fig 8(C) respectively. The clockwise spiral phase distribution indicated the generation of OAM mode +1 and it has been obviously observed from different position as shown in Fig 8(C). Due to the position-by-position scan, there are some dislocations in the measured phase distribution.

### 3.5. Mode purity and SAR analysis

The mode purity of an OAM wave is defined as the ratio of the power in the dominant mode to the total power of the wave. Purity of OAM mode is major parameter in order to analyze and evaluate the state of the OAM waves. In general, generated OAM waves is combination of multiple OAM modes. In this work, the mode purity analysis is performed using the phase distribution obtained from the CST simulation and the final results are obtained based on Fourier transform

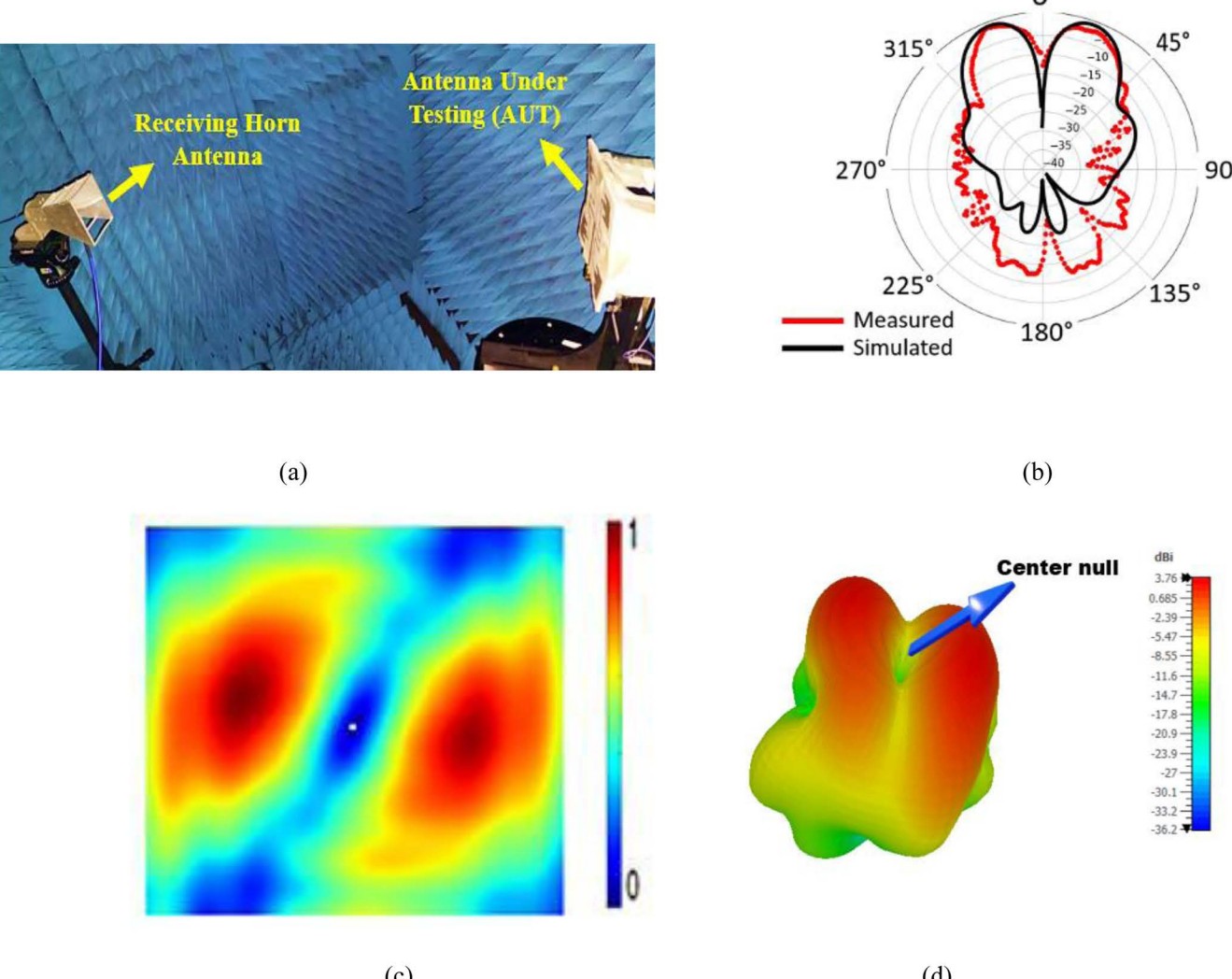

**Fig 7. Far field results at 3.5 GHz (A) measurement setup; (B) Simulated and Measured 2D polar form radiation pattern; (C) Simulated near field intensity distribution; (D) Simulated 3D radiation pattern at 3.5 GHz.**

using Python programming. It can be observed from Fig 9(A) that the OAM mode purity for mode +1 is 52.12%, which is the highest purity compared to the other OAM and the targeted OAM mode dominates the spectrum. The SAR analysis was performed by co-simulating the designed antenna on the Hugo voxel human body at 3.5 GHz with input power 0.5W in the CST MWS. The SAR calculation was performed with antenna placed at 2 mm away from the body considering this 2 mm thickness as the thickness of a cloth to match with practical scenarios. The maximum values of the simulated SAR was 0.00627 W/kg for 1g of tissue as shown in Fig 9(B). For 10g of tissue, the value obtained was 0.00355 W/kg. These results are well below the limits of IEEE C95.1-2005 (1.6 W/kg) and EN 50361-2001 (2.0 W/kg) [29]. The following equation was applied with the input power used to calculate the SAR for the proposed OAM wearable antenna:

$$SAR = \frac{\sigma |E^2|}{\rho} \tag{4}$$

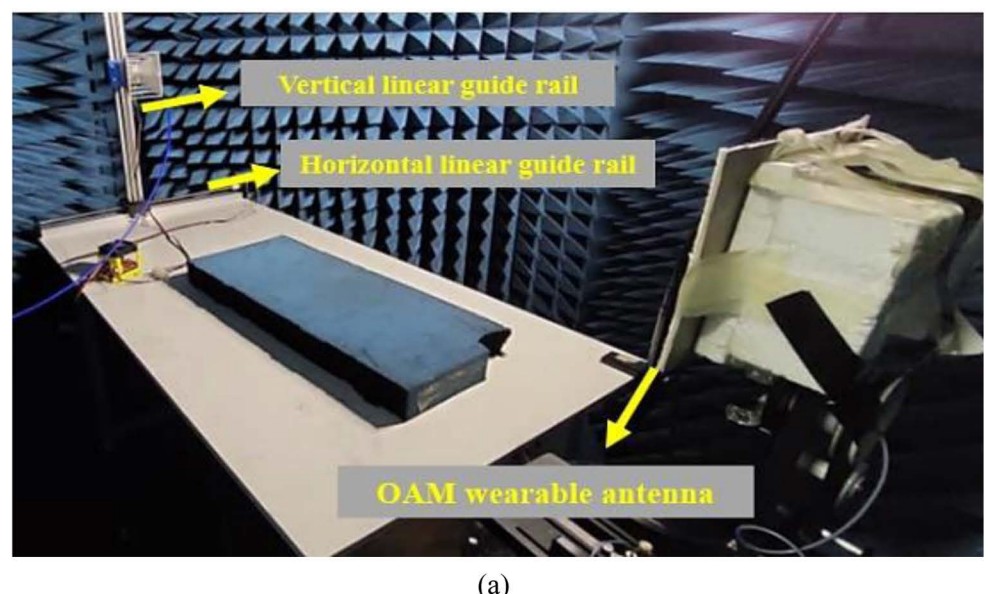

(a)

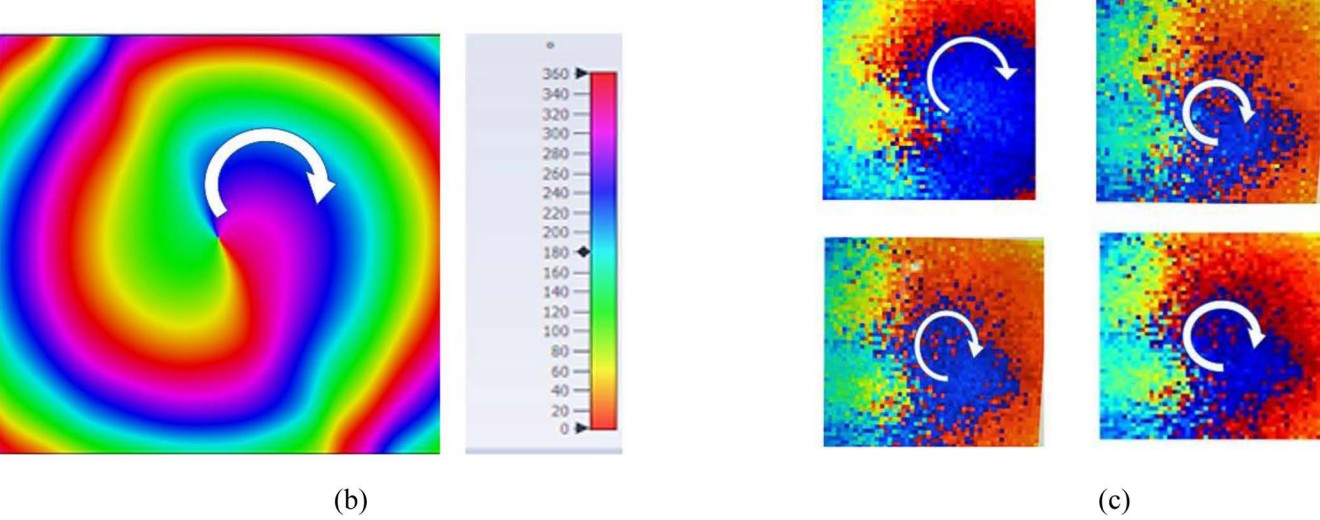

(b)　　　　　　　　　　　　　　　　　　　　(c)

**Fig 8. Phase distribution of the proposed antenna (A) Measurement setup; (B) Simulated phase distribution; (C) Measured phase distribution.**

where ρ is the mass density of the tissue in kg/m³, E is the electric field in V/m, and σ is the conductivity of the tissue in S/m.

### 3.6. Influence of bending on OAM characteristics

Due to its target application for wearable and body-centric communications, antenna deformation (in the form of bending) is expected during normal human activities, and may influence its performance [30]. Hence, the influence of antenna bending on the characteristics of OAM waves was investigated through simulation (0°, 5°, 10°, and 15°). The antenna was bent along the y-axis across the patches' width. This ensures that the resonance frequency does not change as it is sensitive to patch length [7]. The obtained results are tabulated in Table 3. When the proposed antenna was bent from

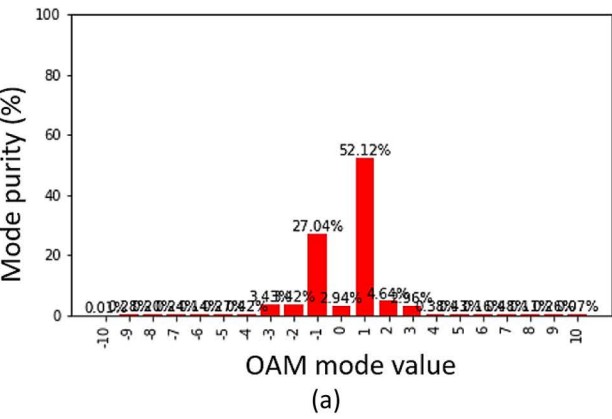

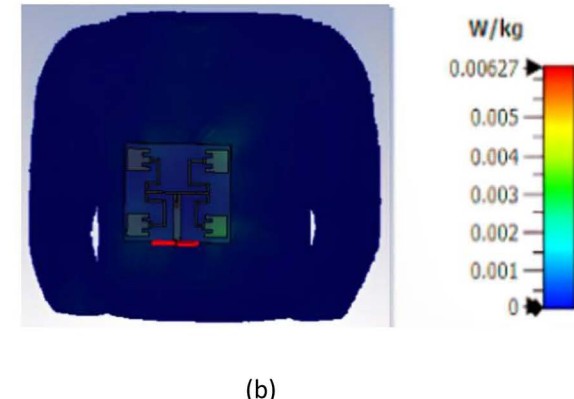

**Fig 9. Simulated analysis (A) Mode purity; (B) SAR.**

**Table 3. Effect of antenna bending on OAM waves characteristics.**

| Bending angle | Intensity distribution | Phase distribution and Mode purity | 2D polar form |
|---|---|---|---|
| 0° | | 52.12% purity | |
| 5° | | 40.36% purity | |
| 10° | | 27.67% purity | |
| 15° | | 2.79% purity | |

0° to a maximum of 15°, the generated mode purity decreased significantly from 52.12% to 2.79%. This is mainly due to the distortion of the spiral phase distribution and vortex core at the centre of intensity distribution. In contrast, when the antenna was flat, a precise null at the boresight direction (0°) was observed in the radiation pattern. As the bending angle increased, the 2D radiation pattern at x-z plane shifted towards the right side, and there was no more null in the boresight direction.

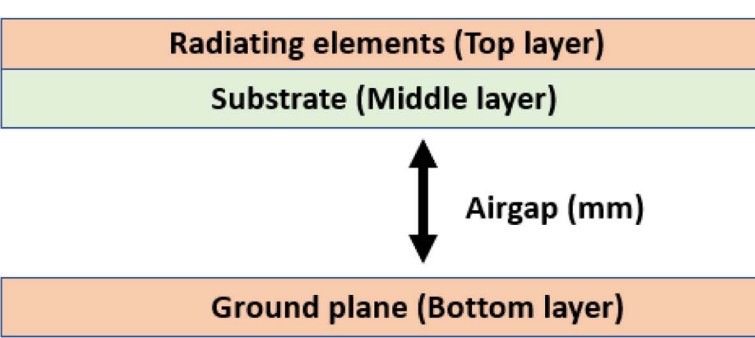

**Fig 10. Implementation of airgap in the proposed antenna.**

(a)

(b)

(c)

**Fig 11. Effect of airgap thickness (A) Mode purity; (B) Gain; (c) Bandwidth.**

**Table 4. Comparison with some of the previous works.**

| Ref | Freq (GHz) / Antenna dimension | Mode | Flexible/ Bending analysis | SAR analysis | Mode purity | Bandwidth (MHz) | Number of elements and ports | Operated at 5G NR bands / Design complexity | OAM analysis |
|---|---|---|---|---|---|---|---|---|---|
| [9] | 10 / No available | +1 | No/ No | No | N/P | $\approx 500$ | 8 and 1 | No / Complex because of eight elements | Phase distribution and 2D polar form |
| [10] | 1.550 / Not available | −1 | No/ No | No | N/P | < 300 | 6 and 1 | No / Complex because of six elements | Phase distribution and 2D polar form |
| [11] | 2.4 / 2.4 $\lambda_0$ × 2.4 $\lambda_0$ | +1, +2 | No/ No | No | N/P | 90 | 8 ($l + 1$) and 2 | No / Complex because of sixteen elements | Phase distribution and 2D polar form |
| [12] | 5.45 / Not available | −1 | No/ No | No | > 90% | < 100 | 8 and 1 | No / Complex because of eight elements | Phase distribution and mode purity |
| [13] | 10 / Not available | −1 | No/ No | No | N/P | 500 MHz | 8 and 1 | No / Simple | Phase distribution |
| [14] | 5.37 to 5.66 / 2.2 $\lambda_0$ × 2 $\lambda_0$ | +1, −1 | No/ No | No | N/P | 300 ($l-1$) 290 ($l+1$) | 4 and 2 | No / Complex because of complicated feeding network | Phase distribution and 2D polar form |
| [15] | 5.8 GHz / Radius is 2 $\lambda_0$ | +1, −1 | No/ No | No | N/P | $\approx 500$ | 15 and 1 | No / Complex because of sixteen elements | Phase distribution and 2D polar form |
| [16] | 3.360 / 0.272 $\lambda_0$ × 0.272 $\lambda_0$ | +1 | No/ No | No | N/P | < 100 | 6 and 1 | Yes / Complex because of six elements and large dimension | Phase and intensity distribution |
| [20] | 3.5 / 2.15 $\lambda_0$ × 1.86 $\lambda_0$ | +1 | No/ No | No | 56.14% | 420 MHz | 4 and 1 | Yes / Simple structure however it is rigid and less bandwidth and gain | 2D polar form and mode purity only |
| **This work** | 3.5 / 1.98 $\lambda_0$ × 1.83 $\lambda_0$ | +1 | Yes/ Yes | Yes | 52.12% | 430 | 4 and 1 | Yes / Simple as only four elements are used | Phase distribution, intensity distribution, 2D polar form and mode purity |

### 3.7. Airgap technique for gain enhancement

Fig 10 illustrates method of inserting airgap within the proposed antenna to increase the gain. As shown in Fig 10, the substrate thickness which is a dielectric material is increased by inserting airgap between the substrate and the ground plane. The thickness of the substrate or dielectric material determines the magnitude of radiation of electromagnetic wave. Hence, if the thickness of the substrate is increased to a certain point, the dielectric constant of the radiating plane as well as the electric field concentration on the lossy epoxy is reduced which eventually enhances the antenna gain [31]. For this paper, the air gap was varied from 0 mm to 2 mm and the effect of airgap on antenna gain, mode purity and bandwidth was analyzed.

From Fig 11(A), it can be observed that, increase in airgap thickness to a certain point increases the OAM mode purity. The OAM mode purity increased to 54.42% from 52.12% when the airgap thickness was increased from 0 mm to 1 mm.

However, there was sharp drop on mode purity when the airgap was increased beyond 1 mm. Fig 11(B) shows the effect of airgap technique on antenna gain. It can be observed that, at 0 mm airgap (no airgap), the antenna gain was 3.762 dBi while the gain increased significantly as the thickness of airgap was increased to 1 mm. The maximum gain obtained was 5.327 dBi at 1 mm of airgap thickness. By increasing the airgap thickness from 0 mm to 1 mm, the gain enhanced by 41.6%. However, the gain dropped slightly from 5.327 dBi to 5.317 dBi when the airgap thickness was further increased. Lastly, as shown in Fig 11(C), increase in airgap thickness reduces the bandwidth of the antenna. Without airgap (0 mm), the bandwidth obtained was 930 MHz while it dropped drastically to 330 MHz when the airgap thickness was 2 mm. Therefore, there is a trade-off between antenna gain and antenna bandwidth when it comes implementation of airgap technique.

Table 4 compares the proposed OAM wearable antenna with previously published works from 1 GHz band to 10 GHz band. It can be observed that the proposed antenna has generated within the 5G NR bands with wider bandwidth compared to the other works. Besides, the number of elements used were less compared to the other works. Moreover, in this paper, all the four characteristics of OAM waves are analysed to confirm the generation of OAM waves using flexible textile antenna at 5G NR bands.

## 4. Conclusion

In this paper, a wearable antenna was designed, fabricated, and tested to generate OAM waves with mode +1. The measured results proved the successful generation of OAM waves using a wearable textile antenna for the very first time to the best of the authors knowledge. Moreover, measurements were carried out to investigate the proposed antenna performance on the human body and different bending conditions. The obtained results indicated that the proposed antenna is suitable for wearable communications. The obtained gain was 3.18 dBi along with a bandwidth coverage of 430 MHz with 52.12% of OAM mode purity at the desired frequency of 5G NR bands. When the OAM antenna was bent from 0° to 15°, the mode purity dropped from 52.12% to 2.79%. Moreover, by inserting 1 mm airgap, the mode purity and gain increased to 54.42% and 5.327 dBi respectively while the bandwidth dropped to 330 MHz. The proposed wearable OAM antenna covered n77 and n78 bands. Besides, the proposed antenna's SAR value is within the safe range for deployment on the human body as it obtained 0.00627 W/kg for 1g of tissue and 0.00355 W/kg for 10g of tissue. Furthermore, a comprehensive investigation was performed to study the influence of bending on OAM wave characteristics. Lastly, an in-depth simulated based study was carried out to propose the use of airgap technique for gain enhancement in wearable textile antennas. The proposed design, which introduces a new feasible method for the OAM waves-based wearable communication systems, is verified by the good agreement between the measurement findings and the simulation results.

## Author contributions

**Conceptualization:** Shehab Khan Noor.

**Data curation:** Shehab Khan Noor.

**Formal analysis:** Shehab Khan Noor.

**Funding acquisition:** Shehab Khan Noor.

**Investigation:** Shehab Khan Noor.

**Methodology:** Shehab Khan Noor, Arif Mawardi Ismail, **Nassrin** I. M. Elamin.

**Software:** Shehab Khan Noor, Arif Mawardi Ismail, Mohd Najib Mohd Yasin, Mohamed Nasrun Osman.

**Supervision: Nassrin** I. M. Elamin, Mohd Najib Mohd Yasin, Ping Jack Soh, Ali H. Rambe.

**Validation:** Nurulazlina Ramli, Adel Y. I. Ashyap.

**Writing – original draft:** Shehab Khan Noor.

**Writing – review & editing:** Arif Mawardi Ismail, **Nassrin** I. M. Elamin, Ping Jack Soh, Nurulazlina Ramli.

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
