## [Decision Letter · Decision Letter 0]

17 Apr 2023

Dear Dr. Elamin,

Thank you for submitting your manuscript to PLOS ONE. After careful consideration, we feel that it has merit but does not fully meet PLOS ONE’s publication criteria as it currently stands. Therefore, we invite you to submit a revised version of the manuscript that addresses the points raised during the review process.

We look forward to receiving your revised manuscript.

Kind regards,

Chan Hwang See, Ph.D.

Academic Editor

PLOS ONE

Journal Requirements:

2. Please note that PLOS ONE has specific guidelines on code sharing for submissions in which author-generated code underpins the findings in the manuscript. In these cases, all author-generated code must be made available without restrictions upon publication of the work. 

Please review our guidelines at https://journals.plos.org/plosone/s/materials-and-software-sharing#loc-sharing-code and ensure that your code is shared in a way that follows best practice and facilitates reproducibility and reuse.

**Additional Editor Comments:**

Based on the comments from both reviewers, this work is required a major revision. Hence, we do encourage you to address the concerns and criticisms of the reviewers detailed at the bottom of this letter and resubmit your article once you have updated it accordingly.

Reviewers' comments:

Reviewer's Responses to Questions

**Comments to the Author**

1. Is the manuscript technically sound, and do the data support the conclusions?

Reviewer #1: Yes

Reviewer #2: Yes

2. Has the statistical analysis been performed appropriately and rigorously?

Reviewer #1: N/A

Reviewer #2: Yes

3. Have the authors made all data underlying the findings in their manuscript fully available?

Reviewer #1: Yes

Reviewer #2: Yes

4. Is the manuscript presented in an intelligible fashion and written in standard English?

Reviewer #1: Yes

Reviewer #2: Yes

Reviewer #1: This article presents a way of designing wearable antenna. It is presented in well organized manner and many issues related to wearable antennas have been addressed. However, following points need to be considered before its acceptance for publications.

1. Almost all figures are too hazy.

2. Review PDF is not properly created, Unable to find the referred figures from the text.

3. Gain value shown here, is measured or simulated? If it is simulated then it has to verified by measured results and vice versa.

4. What about the efficiency of the antenna?

5. There are many wearable antenna available in the market? Where the proposed antenna finds novelty?

6. Pls keep a dimension comparison in the comparison table. Mind that your proposed antenna has a large dimension(170 mm x 156 mm).

7. How the airgap between substrate and ground plane been created physically for increasing gain?

8. How the antenna structure following UCA pattern? mention in the introduction part.

Reviewer #2: This research work has presented and realized a wearable antenna to generate orbital angular momentum waves with mode +1. Promising results have been achieved and well discussed in the well-organized manuscript. So, the results have been experimentally validated and highlighted by providing a fair comparison with state-of-the-art. Although the concept and idea of this work were found interesting and they seem attractive for the scientific society, authors are requested to carefully address the following comments to improve the quality of the manuscript prior to final recommendation.

1) The design process of the proposed wearable textile array antenna can be briefly elaborated in the abstract section. Since this is an array antenna, please explain how authors have addressed the mutual coupling issue between the radiation elements in the proposed structure?

2) Please highlight advantages of the proposed wearable textile array antenna in the abstract section.

3) Introduction section can be improved by adding more comprehensive discussions along with suitable references. For example, mutual coupling reduction is an important issue to realize an array antenna. There are various decoupling methods in the literature which can be briefly elaborated in the introduction section along with proper references. Below are helpful suggestions.

"A Comprehensive Survey on "Various Decoupling Mechanisms with Focus on Metamaterial and Metasurface Principles Applicable to SAR and MIMO Antenna Systems"", IEEE Access, vol. 8, pp. 192965-193004, 2020.

“Study on Isolation and Radiation Behaviours of a 34×34 Array-Antennas Based on SIW and Metasurface Properties for Applications in Terahertz Band Over 125-300 GHz”, Optik, International Journal for Light and Electron Optics, Volume 206, March 2020, 163222.

"Isolation Enhancement of Densely Packed Array Antennas with Periodic MTM-Photonic Bandgap for SAR and MIMO Systems", IET Microwaves, Antennas & Propagation, Volume 14, Issue 3, February 2020, pp. 183 - 188.

"Surface Wave Reduction in Antenna Arrays Using Metasurface Inclusion for MIMO and SAR Systems", Radio Science, 54, 1067–1075, 2019.

"Mutual-Coupling Isolation Using Embedded Metamaterial EM Bandgap Decoupling Slab for Densely Packed Array Antennas", IEEE Access, vol. 7, pp. 5182–51840, April 29, 2019.

"Mutual Coupling Suppression Between Two Closely Placed Microstrip Patches Using EM-Bandgap Metamaterial Fractal Loading", IEEE Access, vol. 7, Page(s): 23606 – 23614, March 5, 2019.

"Interaction Between Closely Packed Array Antenna Elements Using Metasurface for Applications Such as MIMO Systems and Synthetic Aperture Radars", Radio Science, Volume53, Issue11, November 2018, Pages 1368-1381.

“Antenna Mutual Coupling Suppression Over Wideband Using Embedded Periphery Slot for Antenna Arrays”, Electronics, 2018, 7(9), 198.

“Study on Isolation Improvement Between Closely Packed Patch Antenna Arrays Based on Fractal Metamaterial Electromagnetic Bandgap Structures”, IET Microwaves, Antennas & Propagation, Volume 12, Issue 14, 28 November 2018, p. 2241 – 2247.

“Meta-surface Wall Suppression of Mutual Coupling between Microstrip Patch Antenna Arrays for THz-band Applications”, Progress in Electromagnetics Research Letters, Vol. 75, page 105-111, 2018.

4) The array structure is constructed of 4 radiation elements, please explain why authors have implemented two cuts in each radiation element close to their feeding part? All the radiation elements are connected to each other by meander lines, please describe how authors have optimized these meander lines? (from their size point of view and from their topology point of view).

5) The feeding mechanism of the array should be elaborated in depth.

6) Quality of the figures need to be improved.

7) Can authors add the 2D curves of the radiation gain and efficiency before and after vertical and horizontal bending to compare the effect of bending on the radiation properties?

8) In the comparison table 4, please add the term “design complexity” as well. Following paper can be added to this table for its extension.

"Metasurface-Inspired Flexible Wearable MIMO Antenna Array for Wireless Body Area Network Applications and Biomedical Telemetry Devices", IEEE Access, vol. 11, pp. 1039-1056, 2023.

"Metamaterial-Inspired Antenna Array for Application in Microwave Breast Imaging Systems for Tumor Detection", IEEE Access, vol. 8, pp. 174667-174678, 2020, doi: 10.1109/ACCESS.2020.3025672.

**Do you want your identity to be public for this peer review?** For information about this choice, including consent withdrawal, please see our Privacy Policy

Reviewer #1: **Yes: ** Saumya Das

Reviewer #2: No

---

## [Author Response · Author response to Decision Letter 1]

27 Jun 2023

Dear Editor,

Thank you for allowing a resubmission of our manuscript with revised version, with an opportunity to address the reviewers’ comments. We really appreciate your timely handling of our manuscript and the excellent review process. After revising the manuscript based on the Reviewers comments that we have received, we would like to submit the revised manuscript to be considered for publications in PLOS ONE. In the revision, we have addressed all the comments and suggestions given by the Editor and the Reviewers. For convenience, the response to the reviewers are written and it is in bold fonts. The changes made in the revised version of our manuscript are detailed in the point-by point response to the Reviewers comments. Lastly, we would like to thank the Editor and the Reviewers of PLOS ONE, whose efforts have significantly improved the quality of the paper. We are uploading (a) our point-by-point response to the comments (below) (response to reviewers), (b) an updated manuscript with yellow highlighting indicating changes (Revised Manuscript with Track Changes), and (c) a clean updated manuscript without highlights (Manuscript).

---

## [Decision Letter · Decision Letter 1]

16 Aug 2023

Dear Dr. Elamin,

Thank you for submitting your manuscript to PLOS ONE. After careful consideration, we feel that it has merit but does not fully meet PLOS ONE’s publication criteria as it currently stands. Therefore, we invite you to submit a revised version of the manuscript that addresses the points raised during the review process.

We look forward to receiving your revised manuscript.

Kind regards,

Chan Hwang See, Ph.D.

Academic Editor

PLOS ONE

Reviewers' comments:

Reviewer's Responses to Questions

**Comments to the Author**

Reviewer #2: (No Response)

Reviewer #3: (No Response)

Reviewer #4: (No Response)

2. Is the manuscript technically sound, and do the data support the conclusions?

Reviewer #2: Partly

Reviewer #3: Yes

Reviewer #4: Yes

3. Has the statistical analysis been performed appropriately and rigorously?

Reviewer #2: N/A

Reviewer #3: No

Reviewer #4: Yes

4. Have the authors made all data underlying the findings in their manuscript fully available?

Reviewer #2: Yes

Reviewer #3: Yes

Reviewer #4: Yes

5. Is the manuscript presented in an intelligible fashion and written in standard English?

Reviewer #2: Yes

Reviewer #3: Yes

Reviewer #4: No

Reviewer #2: Authors in this paper have investigated and realized a wearable antenna to generate OAM waves with mode +1. The measured results proved the successful generation of OAM waves using a wearable textile antenna for the very first time to the best of the authors knowledge. The proposed design, which introduces a new feasible method for the OAM waves-based wearable communication systems, is verified by the good agreement between the measurement findings and the simulation results. From this reviewer’s point of view, the topic and content of this paper were found interesting. The promising results have been achieved and evaluated in a well-organized manuscript, also an experimental validation was provided. Although this paper seems attractive for readers, authors are requested to address the following comments to improve its quality prior to final recommendation.

1) The proposed novel experimental design approach mentioned in the title should be briefly elaborated in the abstract section.

2) The advantages of the proposed novel experimental design approach can be highlighted in the abstract section.

3) The dimensions of the single elements and array antennas should be mentioned in the abstract section.

4) Metamaterials and metasurfaces are interesting methods to realize high performance antennas, so to improve the quality of the introduction section authors can add a short paragraph including some information about these techniques. Below are helpful suggestions.

"A Comprehensive Survey of "Metamaterial Transmission-Line Based Antennas: Design, Challenges, and Applications"", IEEE Access, vol. 8, pp. 144778-144808, 2020.

"A Comprehensive Survey on Antennas On-Chip Based on Metamaterial, Metasurface, and Substrate Integrated Waveguide Principles for Millimeter-Waves and Terahertz Integrated Circuits and Systems," IEEE Access, vol. 10, pp. 3668-3692, 2022.

"A Comprehensive Survey on "Various Decoupling Mechanisms with Focus on Metamaterial and Metasurface Principles Applicable to SAR and MIMO Antenna Systems"", IEEE Access, vol. 8, pp. 192965-193004, 2020.

“New compact printed leaky‐wave antenna with beam steering”, Microwave and Optical Technology Letters 58 (1), 215-217, 2016.

“Design and Realization of a Frequency Reconfigurable Antenna with Wide, Dual, and Single-Band Operations for Compact Sized Wireless Applications”, Electronics 10 (11), 1321, 2021.

“Dual-Polarized Highly Folded Bowtie Antenna with Slotted Self-Grounded Structure for Sub-6 GHz 5G Applications”, IEEE Transactions on Antennas and Propagation 70 (4), 3028-3033, 2022.

“A new miniature ultra wide band planar microstrip antenna based on the metamaterial transmission line”, 2012 IEEE Asia-Pacific Conference on Applied Electromagnetics (APACE), 293-297, 2012.

“Compact antenna based on a composite right/left‐handed transmission line”, Microwave and Optical Technology Letters 57 (8), 1785-1788, 2015.

“Study on improvement of the performance parameters of a novel 0.41–0.47 THz on-chip antenna based on metasurface concept realized on 50 um GaAs-layer”, Scientific Reports 10 (11034), 1-9, 2020.

“Printed planar patch antennas based on metamaterial”, International Journal of Electronics Letters 2 (1), 37-42, 2014.

“A new planar broadband antenna based on meandered line loops for portable wireless communication devices”, Radio Science 51 (7), 1109-1117, 2016.

“High-isolation antenna array using SIW and realized with a graphene layer for sub-terahertz wireless applications”, Scientific Reports 11 (10218), 1-14, 2021.

“Composite right–left‐handed‐based antenna with wide applications in very‐high frequency–ultra‐high frequency bands for radio transceivers”, IET Microwaves, Antennas & Propagation 9 (15), 1713-1726, 2015.

“Bandwidth extension of planar antennas using embedded slits for reliable multiband RF communications”, AEU-International Journal of Electronics and Communications 70 (7), 910-919, 2016.

5) At both sides of the feeding lines of the radiation patches, authors have made two cutes, please explain the impact of these cutes?

6) The meander lines connecting the feed lines of the upper patches are different from the meander lines connecting the feed line of the bottom patches, please explain why these meander lines connecting the patches to the main feed line do not have the same configuration? In other words, why are they not symmetric?

7) How have authors addressed the mutual coupling between the radiation patches?

8) Looking at the provided figures, it seems authors have bent the array antenna in one direction (vertical direction), did authors have bent it in another direction (horizontal) as well?

9) Besides Fig. 11 (b) please add the 2D radiation efficiency plot as well.

10) Please support the conclusion with more numerical results.

Reviewer #3: The novelty of the paper should be explained more. There are a lot of kinds of antennas suitable for generating OAM waves and enhancing Gain why specifically your design?

2) One of the biggest deficiencies of the paper is that the mathematical background of the antenna design (calculations, antenna theory, principles of the propagation of the electromagnetic field) is missing. The paper must be improved in this case.

3)The introduction does not provide a comprehensive overview of the topic, and there are insufficient references. For this kind of paper, 25 references used in the paper are not sufficient.compare you results with other papers covering OAM ,5G NR Band and Enhansing Gain article like

a) S. K. Noor, A. M. Ismail, M. N. M. Yasin, M. N. Osman, N. Ramli et al., "Generation of OAM waves and analysis of mode purity for 5g sub-6 ghz applications," Computers, Materials & Continua, vol. 74, no.1, pp. 2239–2259, 2023.

b)Illahi, U.; Iqbal, J.; Irfan, M.; Ismail Sulaiman, M.; Khan, M.A.; Rauf, A.; Bari, I.; Abdullah, M.; Muhammad, F.; Nowakowski, G.; et al. A Novel Design and Development of a Strip-Fed Circularly Polarized Rectangular Dielectric Resonator Antenna for 5G NR Sub-6 GHz Band Applications. Sensors 2022, 22, 5531.

4.Gramatical mistake are too many,need to improve english abit specially in Methodology and Result analysis section.

5. The Conclusion part of the paper is too short and the deeper scientific discussion about the achieved results is missing.

Reviewer #4: 1- The entire text of the manuscript should be reviewed in terms of grammar, writing, and spelling.

1. The background of the research should be given in a separate section. Also, use other up-to-date and valid articles. Finally, at the end of this section, a category of existing methods for designing, increasing gain, increasing bandwidth, and reducing dimensions will be presented.

2. For flexible dielectric, it is mandatory to provide SEM images.

3. Although a series of parametric reports have been presented, it is necessary to investigate the effect of changes in the type of dielectric (felt, jeans, cotton, etc.), change in dimensions, and change in dielectric thickness on each of the characteristics of S11, realized gain, radiation patterns at central frequency, and radiation efficiency.

4. Since the proposed antenna is an array antenna, the Sij (insertion losses) characteristic should also be checked. Therefore, report this characteristic and also check all the above parameter reports for this characteristic.

5. It is better to consider a circuit model for the final array antenna. Then, obtain the values of circuit elements with software such as ADS or CST Design Studio. Calculate the S11, realized gain, and insertion losses characteristics for this model. Then, compare these results with the full-wave results. Calculate the percentage of error, sensitivity, and accuracy.

**Do you want your identity to be public for this peer review?** For information about this choice, including consent withdrawal, please see our Privacy Policy

Reviewer #2: No

Reviewer #3: No

Reviewer #4: **Yes: ** Dr. Farzad Khajeh-Khalili

---

## [Author Response · Author response to Decision Letter 2]

25 Sep 2023

Original Manuscript ID: PONE-D-23-04350

Original Article Title: A novel experimental design approach to generating OAM waves and analysis of airgap technique for gain enhancement in flexible and wearable antennas

To: PLOS ONE Editors

Re: Response to reviewers

Dear Editor,

Thank you for allowing a resubmission of our manuscript with revised version, with an opportunity to address the reviewers’ comments. We really appreciate your timely handling of our manuscript and the excellent review process. After revising the manuscript based on the Reviewers comments that we have received, we would like to submit the revised manuscript to be considered for publications in PLOS ONE. In the revision, we have addressed all the comments and suggestions given by the Editor and the Reviewers. For convenience, the response to the reviewers are written and it is in bold fonts. The changes made in the revised version of our manuscript are detailed in the point-by point response to the Reviewers comments. Lastly, we would like to thank the Editor and the Reviewers of PLOS ONE, whose efforts have significantly improved the quality of the paper. We are uploading (a) our point-by-point response to the comments (below) (response to reviewers), (b) an updated manuscript with yellow highlighting indicating changes (Revised Manuscript with Track Changes), and (c) a clean updated manuscript without highlights (Manuscript).

Best regards,

Shehab Khan Noor

---

## [Decision Letter · Decision Letter 2]

4 Apr 2024

Dear Dr. Elamin,

We look forward to receiving your revised manuscript.

Kind regards,

Suhaib Ahmed, Ph.D.

Academic Editor

PLOS ONE

Journal Requirements:

Additional Editor Comments:

Authors are advised to address the queries raised by the reviewers

Reviewers' comments:

Reviewer's Responses to Questions

**Comments to the Author**

Reviewer #2: (No Response)

Reviewer #3: All comments have been addressed

Reviewer #4: All comments have been addressed

Reviewer #5: All comments have been addressed

2. Is the manuscript technically sound, and do the data support the conclusions?

Reviewer #2: Yes

Reviewer #3: Yes

Reviewer #4: Yes

Reviewer #5: Yes

3. Has the statistical analysis been performed appropriately and rigorously?

Reviewer #2: Yes

Reviewer #3: Yes

Reviewer #4: Yes

Reviewer #5: Yes

4. Have the authors made all data underlying the findings in their manuscript fully available?

Reviewer #2: Yes

Reviewer #3: Yes

Reviewer #4: Yes

Reviewer #5: Yes

5. Is the manuscript presented in an intelligible fashion and written in standard English?

Reviewer #2: Yes

Reviewer #3: Yes

Reviewer #4: Yes

Reviewer #5: Yes

Reviewer #2: Authors in this research paper have investigated and implemented a wearable antenna and it was practically tested to generate Orbital Angular Momentum (OAM) waves with mode +1. The measured results proved the successful generation of OAM waves using a wearable textile antenna for the very first time to the best of the authors knowledge. Moreover, measurements were carried out to investigate the proposed antenna performance on the human body and different bending conditions. The proposed design, which introduces a new feasible method for the OAM waves-based wearable communication systems, is verified by the good agreement between the measurement findings and the simulation results. From this reviewer’s point of view, the topic and content of this paper were found interesting. The promising results have been achieved and evaluated in a well-organized manuscript, also an experimental validation was provided. Although this paper seems attractive for readers in scientific society, authors are requested to address the following comments to improve its quality prior to final recommendation.

1) In the abstract section please briefly mention the design process of the proposed wearable textile array antenna?

2) In the abstract section please mention how the isolation between the array elements have been addressed?

3) In the abstract section please mention how much is the distance between the radiation elements?

4) Advantages of the proposed wearable textile array antenna should be highlighted in the abstract section.

5) Introduction section can be improved by adding more discussions on various methods to realize high performance antennas such as metamaterial and metasurface approaches. Below are helpful suggestions.

“A comprehensive survey of" Metamaterial Transmission-Line Based Antennas: Design, Challenges, and Applications"”, IEEE Access 8, 144778-144808, 2020.

"A Comprehensive Survey on Antennas On-Chip Based on Metamaterial, Metasurface, and Substrate Integrated Waveguide Principles for Millimeter-Waves and Terahertz Integrated Circuits and Systems," in IEEE Access, vol. 10, pp. 3668-3692, 2022.

“Design and Realization of a Frequency Reconfigurable Antenna with Wide, Dual, and Single-Band Operations for Compact Sized Wireless Applications”, Electronics 10 (11), 1321, 2021.

“Super-wide impedance bandwidth planar antenna for microwave and millimeter-wave applications”, Sensors 19 (10), 2306, 2019.

“A new miniature ultra wide band planar microstrip antenna based on the metamaterial transmission line”, 2012 IEEE Asia-Pacific Conference on Applied Electromagnetics (APACE), 293-297, 2012.

“Impedance Bandwidth Improvement of a Planar Antenna Based on Metamaterial-Inspired T-Matching Network”, IEEE Access 9, 67916 – 67927, 2021.

“Improved adaptive impedance matching for RF front-end systems of wireless transceivers”, Scientific Reports 10 (14065), 1-11, 2020.

“A Flexible and Pattern Reconfigurable Antenna with Small Dimensions and Simple Layout for Wireless Communication Systems Operating over 1.65–2.51 GHz”, Electronics 10 (5), 601, 2021.

“Modified U-shaped resonator as decoupling structure in MIMO antenna”, Electronics 9 (8), 1321, 2020.

“Compact Quad-Element High-Isolation Wideband MIMO Antenna for mm-Wave Applications”, Electronics 10 (11), 1300, 2021.

“New compact printed leaky‐wave antenna with beam steering”, Microwave and Optical Technology Letters 58 (1), 215-217, 2016.

6) The feeding mechanism of the array antenna should be mentioned in detail.

7) Why have the authors made some cuts at the corner of each patch? The impact of this method should be elaborated in depth.

8) Why are the meandered lines connecting the patches not symmetric?

9) Is it possible to apply the proposed method to implement an array antenna with a higher number of radiation elements?

10) How have authors reduced the mutual coupling between the radiation elements which is an important parameter to realize array antennas? Below are some helpful methods to improve the isolation between the array elements which can be briefly mentioned in the introduction section.

“A comprehensive survey on “Various decoupling mechanisms with focus on metamaterial and metasurface principles applicable to SAR and MIMO antenna systems””, IEEE Access 8, 192965-193004, 2020.

“Study on Isolation and Radiation Behaviours of a 34×34 Array-Antennas Based on SIW and Metasurface Properties for Applications in Terahertz Band Over 125-300 GHz”, Optik, International Journal for Light and Electron Optics, Volume 206, March 2020, 163222.

"Isolation Enhancement of Densely Packed Array Antennas with Periodic MTM-Photonic Bandgap for SAR and MIMO Systems", IET Microwaves, Antennas & Propagation, Volume 14, Issue 3, February 2020, pp. 183 - 188.

"Surface Wave Reduction in Antenna Arrays Using Metasurface Inclusion for MIMO and SAR Systems", Radio Science, 54, 1067–1075, 2019.

"Mutual-Coupling Isolation Using Embedded Metamaterial EM Bandgap Decoupling Slab for Densely Packed Array Antennas", IEEE Access, vol. 7, pp. 5182–51840, April 29, 2019.

"Mutual Coupling Suppression Between Two Closely Placed Microstrip Patches Using EM-Bandgap Metamaterial Fractal Loading", IEEE Access, vol. 7, Page(s): 23606 – 23614, March 5, 2019.

"Interaction Between Closely Packed Array Antenna Elements Using Metasurface for Applications Such as MIMO Systems and Synthetic Aperture Radars", Radio Science, Volume53, Issue11, November 2018, Pages 1368-1381.

“Antenna Mutual Coupling Suppression Over Wideband Using Embedded Periphery Slot for Antenna Arrays”, Electronics, 2018, 7(9), 198.

“Study on Isolation Improvement Between Closely Packed Patch Antenna Arrays Based on Fractal Metamaterial Electromagnetic Bandgap Structures”, IET Microwaves, Antennas & Propagation, Volume 12, Issue 14, 28 November 2018, p. 2241 – 2247.

“Meta-surface Wall Suppression of Mutual Coupling between Microstrip Patch Antenna Arrays for THz-band Applications”, Progress in Electromagnetics Research Letters, Vol. 75, page 105-111, 2018.

Reviewer #3: Author incorporates all the suggested reviews and now I believe it’s ready to accept.they also update the grammatical mistakes too.

Reviewer #4: This manuscript has found suitable conditions compared to the previous version. Therefore, it can be published in this magazine. However, it is recommended that authors carefully review the entire text of the manuscript for spelling and grammar.

Reviewer #5: Authors have addressed most of the queries raised by various reviewers in appreciable manner.

However, there are few queries from my end and hope authors shall address them. The

manuscript can be accepted for publication with minor revision.

Comments:

#1. There is no mention about the polarization of the proposed design in the manuscript.

#2. There are several grammatical errors in the manuscript.

#3. The authors in the manuscript don’t mention whether the results like radiation pattern, gain,

bandwidth etc. achieved are on-body or free space results. And if there is any difference

between the two the authors are encouraged to showcase the same.

#4. Is there any effect of body on the generation of OAM waves when the antenna is mounted on

the body? And if yes, discuss in the manuscript

**Do you want your identity to be public for this peer review?** For information about this choice, including consent withdrawal, please see our Privacy Policy

Reviewer #2: No

Reviewer #3: **Yes**

Reviewer #4: **Yes**

Reviewer #5: No

---

## [Author Response · Author response to Decision Letter 3]

14 Apr 2024

Thank you for allowing a resubmission of our manuscript with revised version, with an opportunity to address the reviewers’ comments. We really appreciate your timely handling of our manuscript and the excellent review process. After revising the manuscript based on the Reviewers comments that we have received, we would like to submit the revised manuscript to be considered for publications in PLOS ONE. In the revision, we have addressed all the comments and suggestions given by the Editor and the Reviewers. For convenience, the response to the reviewers are written and it is in bold fonts. The changes made in the revised version of our manuscript are detailed in the point-by point response to the Reviewers comments. Lastly, we would like to thank the Editor and the Reviewers of PLOS ONE, whose efforts have significantly improved the quality of the paper. We are uploading (a) our point-by-point response to the comments (below) (response to reviewers), (b) an updated manuscript with yellow highlighting indicating changes (Revised Manuscript with Track Changes), and (c) a clean updated manuscript without highlights (Manuscript).

---

## [Decision Letter · Decision Letter 3]

25 Jul 2024

A novel experimental design approach to generating orbital angular momentum waves using wearable textile antenna for sub-6 GHz 5G

PONE-D-23-04350R3

Dear Dr. Elamin,

We’re pleased to inform you that your manuscript has been judged scientifically suitable for publication and will be formally accepted for publication once it meets all outstanding technical requirements.

Kind regards,

Suhaib Ahmed, Ph.D.

Academic Editor

PLOS ONE

Additional Editor Comments (optional):

Reviewers have accepted the changes done by the authors in their revised manuscript.

Reviewers' comments:

Reviewer's Responses to Questions

**Comments to the Author**

Reviewer #4: All comments have been addressed

Reviewer #5: All comments have been addressed

2. Is the manuscript technically sound, and do the data support the conclusions?

Reviewer #4: Yes

Reviewer #5: Yes

3. Has the statistical analysis been performed appropriately and rigorously?

Reviewer #4: Yes

Reviewer #5: Yes

4. Have the authors made all data underlying the findings in their manuscript fully available?

Reviewer #4: Yes

Reviewer #5: Yes

5. Is the manuscript presented in an intelligible fashion and written in standard English?

Reviewer #4: Yes

Reviewer #5: Yes

Reviewer #4: After revising this manuscript, the previous errors have been fixed and this work has found a better quality.

According to the previous editions, this work can now be published in this journal.

Reviewer #5: Authors have addressed most of the queries raised by me in an appreciable manner. So, I recommend this for publication.

**Do you want your identity to be public for this peer review?** For information about this choice, including consent withdrawal, please see our Privacy Policy

Reviewer #4: **Yes: ** Farzad Khajeh-Khalili

Reviewer #5: **Yes: ** Umhara Rasool Khan

---

## [Editor Report · Acceptance letter]

PONE-D-23-04350R3

PLOS ONE

Dear Dr. Elamin,

I'm pleased to inform you that your manuscript has been deemed suitable for publication in PLOS ONE. Congratulations! Your manuscript is now being handed over to our production team.

Kind regards,

on behalf of

Dr. Suhaib Ahmed

Academic Editor

PLOS ONE